# A randomised experiment of health, cost and social norm message frames to encourage acceptance of swaps in a simulation online supermarket

**Amanda Bunten**[1], **Lucy Porter**[1]*, **Jet G. Sanders**[1,2], **Anna Sallis**[1], **Sarah Payne Riches**[1,3], **Paul Van Schaik**[4], **Marta González-Iraizoz**[5], **Tim Chadborn**[1], **Suzanna Forwood**[6]

**1** Public Health England Behavioural Insights, London, United Kingdom, **2** Department of Psychological and Behavioural Science, London School of Economics and Political Science, London, United Kingdom, **3** Nuffield Department of Primary Care Health Sciences, University of Oxford, Oxford, United Kingdom, **4** School of Social Sciences, Humanities and Law, Teesside University, Middlesbrough, United Kingdom, **5** Warwick Business School, University of Warwick, Coventry, United Kingdom, **6** School of Psyhology and Sport Science, Anglia Ruskin University, Cambridge, United Kingdom

* L.Porter@exeter.ac.uk

## Abstract

Offering lower-energy food swaps to customers of online supermarkets could help to decrease energy (kcal) purchased and consumed. However, acceptance rates of such food swaps tend to be low. This study aimed to see whether framing lower-energy food swaps in terms of cost savings or social norms could improve likelihood of acceptance relative to framing swaps in terms of health benefits. Participants ($n$ = 900) were asked to shop from a 12-item shopping list in a simulation online supermarket. When a target high-energy food was identified in the shopping basket at check-out, one or two lower-energy foods would be suggested as an alternative (a "swap"). Participants were randomised to only see messages emphasising health benefits (fewer calories), cost benefits (lower price) or social norms (others preferred this product). Data were analysed for 713 participants after exclusions. Participants were offered a mean of 3.17 swaps ($SD$ = 1.50), and 12.91% of swaps were accepted (health = 14.31%, cost = 11.49%, social norms = 13.18%). Swap acceptance was not influenced by the specific swap frame used (all $p$ > .170). Age was significantly and positively associated with swap acceptance ($b$ = 0.02, $SE$ = 0.00, $p$ < .001), but was also associated with smaller decreases in energy change ($b$ = 0.46, $SE$ = .19, $p$ = .014). Overall, offering swaps reduced both energy (kcal) per product ($b$ = -9.69, $SE$ = 4.07, $p$ = .017) and energy (kcal) per shopping basket ($t_{712}$ = 11.09, $p$ < .001) from pre- to post-intervention. Offering lower-energy food swaps could be a successful strategy for reducing energy purchased by customers of online supermarkets. Future research should explore alternative solutions for increasing acceptance rates of such swaps.

**Data Availability Statement:** Datasets have been uploaded to the submission as Supporting Information files.

**Funding:** This work was completed as part of the usual business activities of Public Health England and received no additional funding. LP, JGS and MGI were on placement at Public Health England as part of the RCUK internship scheme for part of their involvement in this work.

**Competing interests:** The authors have declared that no competing interests exist.

# Introduction

Overweight and obesity is now the norm in the UK; at the latest estimate, 66% of men and 57% of women had overweight or obesity [1]. Obesity is linked to an increased risk of diseases such as cardiovascular disease and cancer [2], is associated with increased all-cause mortality [3], and significantly reduces individuals' quality-adjusted life years [4].

One of the primary causes of weight gain is an imbalance in the amount of energy consumed against the amount of energy expended through activity and basal metabolic rate [5–7], with other factors such as body weight and physical fitness influencing this relationship [8]. Excess energy intake is facilitated when energy-dense foods are highly visible, widely available and easily accessible, in what some have termed the "obesogenic" environments of modern society [9]. Making small changes to the environments in which people choose and consume foods could be a viable strategy to reduce energy intake [10]. Examples of specific techniques to do this include manipulating the proximity of certain food items in shops and cafeterias [11] or priming consumers with healthy eating messages in supermarkets [12, 13].

Supermarkets are a crucial environment for interventions as a large amount of food is purchased from these outlets; throughout 2019, monthly food retail sales topped £12.5 billion in the UK [14] and many consumers are now using supermarket websites for their weekly shop, with approximately 45% of British shoppers doing at least some of their grocery shopping online in 2018 [15]. Demand for online grocery shopping rose sharply around the world due to the COVID-19 pandemic, with one in five UK families doing some grocery shopping online in 2020 [16] and the majority of consumers in Germany, France, Spain, Italy, the Netherlands and Sweden increasing their purchases of online groceries [17]. Together with evidence from systematic reviews showing that strategies such as manipulating price, availability and visibility of healthier options are effective at encouraging healthier food purchases [18, 19], this points towards the supermarket as an opportune target for intervention.

Online supermarket environments offer the opportunity for tailored interventions based on the products already in consumers' shopping baskets. Some researchers have explored the strategy of offering "swaps" just prior to purchase, by suggesting healthier alternatives to chosen products that are higher in salt, fat, energy or sugar [20, 21]. One study adapted a real online supermarket so that when participants selected foods containing more than 1% saturated fat, they were prompted at the checkout to choose a similar food that was lower in saturated fat (e.g. low-cholesterol margarine in exchange for butter). This simple intervention significantly reduced the saturated fat purchased by consumers [20]. More recent work found that even when the healthier alternatives are dissimilar to the original product (in order to achieve larger decreases in salt content), acceptability of the intervention was maintained [22].

However, the success of this approach depends on consumers accepting the suggested swaps. One study offered participants lower-energy swaps in simulation online supermarket and found that the average reduction in energy purchased (24 kJ) was limited by low acceptance of swaps (a median of one in four swaps was accepted) [21]. More recently, while suggesting food swaps was found to be a successful strategy for reducing the saturated fat purchased by consumers in a simulation online supermarket, efficacy was limited by low acceptance rates (a median of 14% for the group who received this intervention alone) [23].

Neither of these studies explored why swap acceptance rates were low, however Forwood and colleagues suggested that the way that swaps were framed in their study (i.e., in terms of reducing calories) could have been off-putting [21]. Health messages are sometimes perceived to restrict freedom and personal autonomy, leading to psychological reactance and behavioural resistance [24, 25]. In addition, products that are perceived to be healthy are often also perceived to be less palatable than less heathy food products [26]. Research has long shown

that many consumers place lower importance on health messages, and higher importance on taste and price [27], with those who do not prioritise health consuming a less healthy diet [28]. This means that health messages are more likely to appeal to those who are already engaged with health promotion, [12] thus exacerbating existing dietary inequalities for example in fruit and vegetable consumption [29]. Forwood and colleagues suggested that focusing on other product benefits (e.g., cost savings or popularity) could be a strategy to improve swap acceptability [21].

The cost of products has been found to be a particularly crucial factor for those on lower incomes [30–33] and a study investigating consumer choice in online supermarket settings found that lower price can encourage choices of healthier products more effectively than health-status labels [34]. It is therefore possible that highlighting the financial benefits of food swaps could increase the likelihood that consumers accept food swaps compared to messages promoting the health benefits of swaps, particularly for those on lower incomes.

Another highly influential factor in behaviour is the behaviour of other people [35, 36]. The provision of social norms information (describing what other people are doing or approve of) influences both food selection and intake [35, 37], with individuals being more likely to select healthy foods if they believe others have done so previously [38]. Social norms information has been found to be more effective than health information in guiding healthy eating [39], making it likely that a social norms frame could help to encourage consumers in accepting a suggested food swap.

The primary aim of the current study is to test the hypothesis that framing the benefits of lower-energy food swaps in terms of (i) cost savings or (ii) social norms (popularity) will increase swap acceptance at the checkout of a simulation online supermarket, and reduce total energy (kcal) purchased compared to framing the benefits in terms of health (e.g., calories saved). It was also hypothesised that the advantage of framing swaps in terms of cost savings over health benefits would be greater for individuals on low incomes. The experiment was pre-registered at ISRCTN67116897.

## Materials and methods

### Participants and design

This study used a between-subject design, testing the effect of message frame (health frame, social norms frame or cost frame) on acceptance of lower-energy food swaps and the energy content of participants' purchases.

The target sample size was determined with an aim to power the study to detect a minimum increase in swap acceptance from an expected baseline of 25% in the health frame condition [21] up to 35% in the social norms frame and cost frame conditions. In addition, socioeconomic status was estimated to moderate this effect by ± 12% between the lowest SES quartile and the highest. The calculation showed that 900 participants would be required to achieve 80% power.

Participants were recruited through a research agency (Research Now, subsequently renamed Dynata) with the aim of obtaining a representative sample that evenly covered SES quartiles, genders, ages and self-reported BMI scores (range 18 to 40). In addition, screening questions were used to ensure all participants were responsible for at least half of the food/grocery shopping in their household to ensure that the sample would be representative of grocery shoppers in the UK. Finally, participants had to pass an attention check (two multiple choice questions with a range of plausible and implausible answers).

Participants were randomised to see one type of message alongside swap offers (health, cost or social norms). Randomisation was carried out on a 1:1:1 ratio by the online questionnaire

platform (Qualtrics). Participants were blind to their allocation throughout the study, and researchers were blind to participant allocation until analysis. Ethical approval for this study was granted by the University of Cambridge Psychology Research Ethics Committee in September 2015 (reference number PRE.2015.056).

## Materials and measures

**Pre-study questionnaire.**   Participants were asked to provide data on their age in years, gender (male or female), and highest educational qualification (labelled 1–6 from "None", "Up to 4 GCSEs", "5 or more GCSEs", "2 or more A-Levels", "Bachelor's degree", to "Post-graduate degree"). Participants were also asked to indicate how often they purchased a range of food items over the past year ("Not in the last year", "1–3 times in the last year", "4–11 times in the last year", "1–3 times a month", "Once a week", "2–4 times a week", "5 or more times a week"). Finally, they were asked about their responsibility for grocery shopping in their household.

**Online shopping task.**   After the pre-study questionnaire, participants were given a link which took them to a simulation online supermarket (www.woodssupermarket.co.uk). Participants were made aware that the website was not a real commercial site and they would not be required to spend any money. In keeping with the observation that most people use a shopping list when doing their grocery shopping [40], participants were provided with a pre-set list of 12 foods to shop for (S1 File), as per previous research [21]. The list contained both targeted (e.g., bread) and open-ended (e.g., a snack to enjoy now). Participants were advised to choose products that they would normally purchase or have purchased in the past and were also advised to stick to a guideline budget of £25 in total.

An automated algorithm reviewed all items chosen and determined whether potential lower-energy alternative products existed. Swaps were offered at the check-out when an alternative food could be identified that (i) was from the same product category as defined by the shelf location used by the retailer from which the product database came (e.g., Sweet Biscuits, Cheddar Cheese, Fresh Soup; S2 File) as the originally selected food, (ii) weighed between 90–110% of the weight of the originally chosen food, (iii) cost less than the originally selected food by a maximum of 20% (no minimum difference limit was imposed), and (iv) was less energy-dense by at least 100 kJ (24 kcal) per 100g. The algorithm did not incorporate other characteristics (such as brand matching). Where more than one healthier alternative product was identified, swap offers presented participants with two alternatives; otherwise only one alternative was offered.

When a swap was identified, a pop-up appeared at the check-out which presented a thumbnail image of the originally-chosen product on the left and the suggested alternative(s) displayed on the right. Where two healthier alternatives were offered, these were shown simultaneously. Product names and prices were displayed. At the top of the pop-up window, a message was shown highlighting one of (i) the health benefits, (ii) cost benefits or (iii) social norms of choosing the suggested alternative, depending on the condition participants had been allocated to (Fig 1). Participants could either accept (one of) the healthier alternative(s) as a replacement for their original choice product or reject the swap and retain their original choice product. Swaps were offered at the check-out and not the point-of-selection (as has been found to increase swap acceptance rates in previous studies [21]) due to the possibility that accepting a healthier swap may have subsequently led participants to engage in compensatory behaviours. Participants were offered a swap for each swap-relevant product in their basket. The online shopping task was closely modelled on the procedure used by Forwood and colleagues [21], including the provision of a budget and shopping list, and the usage of the same swap-identifying algorithm.

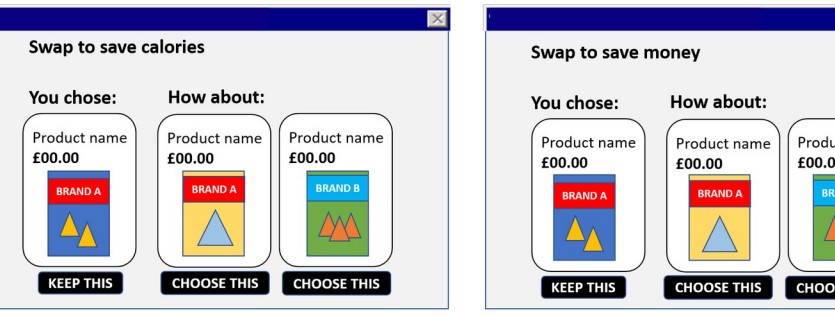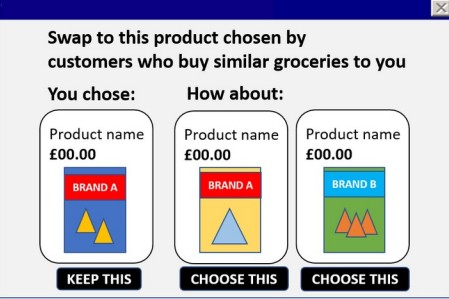

**Fig 1. Mock-up of swap offer pop-ups with message framing.**

**Post-study questionnaire.** Upon completing the shopping task, participants were asked to complete the Post-Study Questionnaire. Participants were asked to rate the online shopping experience out of the following options: "Excellent", "Very Good", "Good", "Fair" and "Poor" (this was coded from 1 to 5 for analysis, with higher scores indicating more positive ratings). They were also asked to rate the acceptability of the swap-suggestion intervention from "Very strongly like", "Strongly like", "Somewhat like", "Indifferent", "Somewhat dislike", "Strongly dislike" and "Very strongly dislike" (coded 1 to 7 for analysis, with higher scores indicating more positive ratings), with the additional option of "I did not notice any alternatives being offered to me". Finally, participants were asked to answer questions assessing how frequently they shopped online for food and non-food items, their weight, height and gross annual household income (labelled 1–4 from "Less than £15,500", "Between £15,500 and £24,999", "Between £25,000 and £39,999" to "More than £40,000").

## Outcomes

The primary outcome measure was whether or not swap offers were accepted (defined as occurring when a healthier alternative was accepted to replace the original choice in their shopping basket). The secondary outcome measures were (i) the energy content (kcal) of individual products involved in the swaps and (ii) the energy content (kcal) of the total shopping baskets.

## Analyses

**Preliminary analyses.** The data were checked and cleaned for duplicate datasets (participants were able to return to re-take the task if they needed to). In these cases, only data from the final visit were retained. Chi-square tests and ANOVAs were performed to check whether the three experimental groups were well matched for baseline characteristics.

**Primary outcome.** Swap acceptance was analysed using binary logistic regression using the lme4 package in R. The unadjusted model analysed the impact of swap frame only, while the adjusted model added age, gender, BMI, education level, income category and the interaction between swap frame and income category. As two models were analysed, the critical significance level was adjusted using a Bonferroni correction to .025. Both models controlled for the number of lower-energy alternatives offered per swap (either one or two were offered each time), and the number of swaps offered to each participant.

For the adjusted model, multilevel analysis (with swaps nested within participants) was planned; however, a fixed-effect model was ultimately conducted as the planned random-

effects model failed to converge. A number of attempts were made to rectify this (such as exploring multicollinearity between predictor variables) and the model converged only once the random effects (i.e., nesting within participant) had been removed from the model.

**Secondary outcomes.** Reductions in energy content per individual swap offer were analysed using multilevel linear regression to assess (i) whether offering swaps significantly reduced the energy content of participants' purchases (original choices versus final purchases), (ii) whether any reduction was moderated by swap frame and (iii) what the potential impact of the offering swaps would have been, were all swaps accepted (original choices versus lower-energy alternatives, regardless of whether they were accepted or not) and (iv) what the potential additional impact of offering swaps would have been, had all swaps been accepted (final purchases versus lower-energy alternatives). As before, swaps were nested within participants, and models controlled for the number of lower-energy alternatives offered per swap, and the number of swaps offered to each participant. Time was accounted for as a dummy variable (e.g., original choices vs. final purchases). As two analyses were conducted for each of the outcomes (observed energy change—analyses i and ii; potential energy change—analyses iii and iv), the critical significance level was adjusted using a Bonferroni correction to .025.

Reductions in energy content per total shopping basket were analysed using t-tests to assess the impact of offering swaps on total purchased energy, and the potential impact of offering swaps had all swaps been accepted. Change in energy content of shopping baskets from pre- to post-swap offer was assessed, and linear regression was used to see whether this differed by swap frame group. An additional model explored whether demographic variables (age, gender, BMI, education level, income category) influenced basket energy change. As three analyses were performed on observed change, the critical p value was adjusted using a Bonferroni correction to .017. Finally, t-tests were used to assess the effect of offering swaps on total basket saturated fat, sugar and salt (all in grams). As three comparisons were conducted on these nutrients, the critical significance value was adjusted using a Bonferroni correction to .017.

# Results

## Preliminary analyses

Participants ($n = 900$, 473 female) aged 18 to 97 ($M = 47.00$, $SD = 16.21$) were randomly allocated to the health frame condition (control; $n = 302$), the cost frame condition ($n = 300$), or the social norms frame condition ($n = 298$). Participants were exposed to one message type only. Of the 900 participants who passed the shopping responsibility and attention checks, 187 were excluded (reasons for exclusion: (i) bought fewer than 10 products [$n = 113$], (ii) bought more than two "off-list" products [$n = 20$], (iii) BMI lower than 18 [$n = 22$], (iv) missing demographic data [$n = 3$] or (v) did not choose any products eligible for swap offers [$n = 17$]), leaving a final sample of 713 participants. The baseline characteristic analysis revealed that excluded participants did not differ from included participants according to gender, education level, income level or swap frame condition (all $p > .610$), however included participants were significantly older ($M = 47.72$, $SD = 15.89$) than excluded participants ($M = 44.23$, $SD = 17.12$, $t_{895} = -2.61$, $p = .009$). Given the large sample size and the small numerical difference between groups, this difference was not considered to be meaningful. Further randomisation checks revealed that the groups (post-exclusions) were well balanced with regards to age, gender, education, income, BMI and energy content (kcal) of their original shopping baskets (Table 1).

In total, 2262 swap offers occurred, with 726 offers showing one alternative product and 1516 offers showing two alternative products. These swap offers were prompted by 606 individual products, for which 228 were associated with one alternative product, and 378 were associated with two alternative products. Individual participants experienced between one and

**Table 1. Randomisation checks and distribution of baseline and demographic characteristics between groups.**

| Factor | Condition | | |
|---|---|---|---|
| | Health $n = 234$ | Cost $n = 238$ | Social Norm $n = 241$ |
| Age in years M (SD) | 47.71 (15.55) | 48.16 (15.62) | 47.29 (16.53) |
| Gender–n (%) Female | 119 (50.9) | 126 (52.9) | 129 (53.5) |
| Education–n (%) | | | |
| None | 7 (3.0) | 6 (2.5) | 5 (2.1) |
| Up to 4 GCSEs | 24 (10.3) | 37 (15.5) | 37 (15.4) |
| 5+ GCSEs | 47 (20.1) | 39 (16.4) | 42 (17.4) |
| 2+ A Levels | 56 (23.9) | 49 (21.0) | 53 (22.0) |
| Bachelor's degree | 64 (27.4) | 67 (28.2) | 72 (29.9) |
| Post-graduate degree | 36 (15.4) | 40 (16.8) | 32 (13.3) |
| Income–n (%) | | | |
| Under £15,500 | 42 (17.9) | 41 (17.2) | 35 (14.5) |
| £15,500 - £24,999 | 48 (20.5) | 48 (20.2) | 57 (23.7) |
| £25,000 - £39,999 | 67 (28.6) | 72 (30.3) | 67 (27.8) |
| £40,000 + | 77 (32.9) | 77 (32.4) | 82 (34.0) |
| BMI M (SD) | 26.97 (7.04) | 27.49 (6.10) | 27.06 (5.86) |
| Baseline total basket energy content (kcal) M (SD) | 3423.09 (728.39) | 3367.37 (528.00) | 3450.23 (506.23) |

14 swap offers each ($M = 3.17$, $SD = 1.50$). Swap offers were spread evenly across the three conditions (health frame = 741 offers, cost frame = 740 offers, social norm frame = 812 offers). Across all groups, 292 swap offers (12.91% of total) were accepted by a total of 202 participants (28.3% of total sample). In the health frame condition 106 swaps (14.31%) were accepted by 78 participants (33.3% of this group), in the cost frame condition 85 swaps (11.49%) were accepted by 58 participants (24.4% of this group), and in the social norms frame condition 107 swaps (13.18%) were accepted by 66 participants (27.4% of this group; Fig 2). On average, each swap offer led to a mean reduction of 9.69 kcal ($SD = 35.14$) in energy purchased (comparing kcal of original choices against final purchases, regardless of whether the swap was accepted or not). The potential average energy reduction (i.e., if all swaps had been accepted, comparing kcal of original choices against alternative products) was 92.93 kcal ($SD = 96.62$). Participants spent between 2.05 and 152.83 minutes on the supermarket task ($M = 19.51$, $SD = 14.46$).

## Main analyses

**Primary outcome.** The unadjusted model found no significant effect of message framing on swap acceptance. With the health frame condition as a baseline, framing the benefits in terms of Cost ($b = 0.19$, $SE = 0.25$, $z = 0.75$, $p = .455$) or Social Norms ($b = 0.34$, $SE = 0.25$, $z = 1.36$, $p = .175$) did not significantly increase the likelihood of accepting a swap. For the adjusted model, no significant main effects or interactions were observed for the independent variable of swap frame condition (see Table 2). Age was significantly associated with swap acceptance, demonstrating a small positive effect on the likelihood of accepting a healthier alternative ($b = 0.02$, $SE = 0.00$, $z = 3.61$, $p < .001$).

**Secondary outcomes.** Multilevel linear regression comparing originally chosen products against finally purchased products revealed that offering swaps significantly reduced the calorie content from participants' original choices to their final purchases ($b = -9.69$, $SE = 4.07$, $t_{3712.086} = -2.38$, $p = .017$) across all intervention groups. Comparing originally chosen products

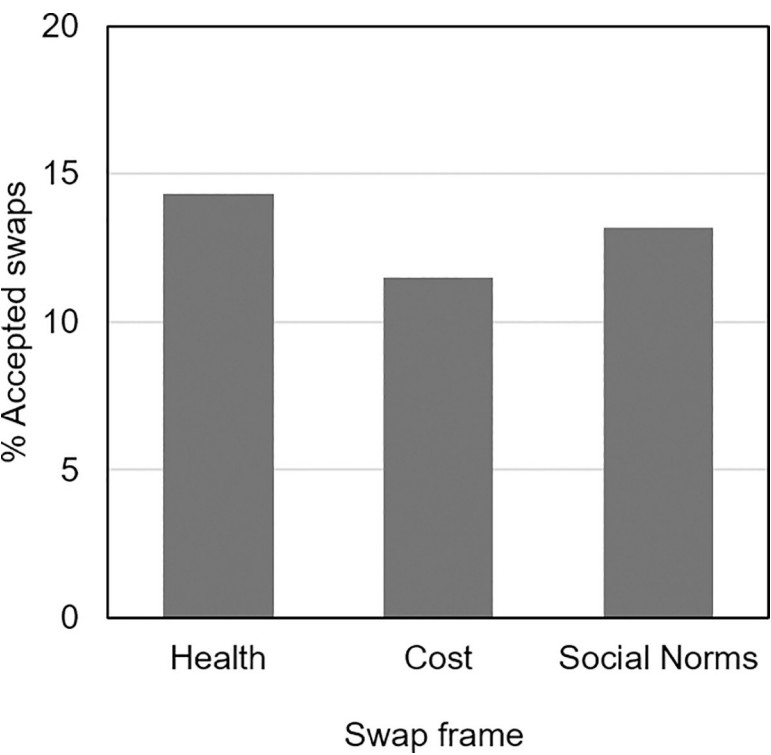

**Fig 2. Percentage of swap offers accepted by swap frame group.**

against the healthier alternative products (i.e., regardless of whether they were accepted or not) revealed that the potential impact of offering swaps was almost ten times greater than that observed ($b$ = -92.93, $SE$ = 4.35, $t_{3724.133}$ = -21.37, $p < .001$). Unsurprisingly, the difference between final products purchased and healthier alternatives was also significant ($b$ = -83.24, $SE$ = 4.39, $t_{3719.167}$ = -18.96, $p < .001$), showing that there was still greater scope for offering swaps to reduce energy purchased. The difference between original products and final purchases was not moderated by either the cost frame ($b$ = 2.45, $SE$ = 10.06, $t$ = 0.24, $p$ = .808) or the social norms frame ($b$ = -0.64, $SE$ = 9.92, $t$ = -0.06, $p$ = .949) relative to the health frame.

Analysis of the total energy content of participants' whole shopping baskets revealed that the average reduction of 30.48 kcal between original choices ($M$ = 3622.85 kcal, $SD$ = 747.80) and final purchases ($M$ = 3594.68, $SD$ = 748.64) was significant ($t_{712}$ = 9.28, $p < .001$). The potential impact of offering swaps (i.e., the difference between original choices and lower-energy alternatives [$M$ = 3127.88 kcal, $SD$ = 585.03]) stood at 494.97 kcal and was also significant ($t_{712}$ = 24.84, $p < .001$). Regressions investigating swap frame effects on change in shopping basket energy content (from original choices to final purchases) revealed no significant effects of the cost or social norms frames relative to the health frame (both $p > .180$). This did not change when the model was adjusted for the impact of demographic variables. The adjusted model revealed that women showed a significantly larger decrease in the energy content of their shopping baskets ($b$ = -16.57, $SE$ = 6.41, $t$ = -2.59, $p$ = .010). No other significant effects were observed. The full results of these models are reported in S1 Table. Analyses testing changes in the total shopping basket content of other nutrients revealed a significant reduction in grams of saturated fat ($M_{Change}$ = -1.09, $SD$ = 5.52, $t_{712}$ = -5.27, $p < .001$), but no significant changes in grams of sugar ($M_{Change}$ = 0.14, $SD$ = 6.12, $t_{712}$ = 0.59, $p$ = .555) or grams of salt ($M_{Change}$ = 0.28, $SD$ = 6.89, $t_{712}$ = 1.08, $p$ = .281).

**Table 2. Model summaries for the main analyses on the outcome of swap acceptance.**

| | *b* (*SE*) | *Z* | *p* |
|---|---|---|---|
| Unadjusted model | | | |
|  Number of lower-energy alternatives in swap[a] | 0.01 (0.16) | 0.04 | .969 |
|  Number of swaps per participant (c) | 0.07 (0.07) | 0.93 | .352 |
|  Swap frame (health frame as baseline) | | | |
|   Social norm frame | 0.19 (0.25) | 0.75 | .455 |
|   Cost frame | 0.34 (0.25) | 1.36 | .175 |
| Adjusted model | | | |
|  Number of lower-energy alternatives in swap[a] | 0.03 (0.14) | 0.22 | .828 |
|  Number of swaps per participant[a] | 0.09 (0.05) | 2.07 | .039 |
|  Swap frame (health frame as baseline) | | | |
|   Social norm frame | -0.06 (0.36) | -0.16 | .874 |
|   Cost frame | -0.11 (0.34) | -0.33 | .743 |
|  Age (years) | 0.02 (0.00) | 3.61 | < .001 |
|  Gender (Female as baseline) | 0.06 (0.14) | 0.45 | .652 |
|  BMI score | -0.00 (0.01) | -0.06 | .956 |
|  Education (Bachelor's degree as baseline) | | | |
|   None | -0.73 (0.39) | -1.86 | .063 |
|   4 GCSEs or fewer | -0.41 (0.20) | -1.99 | .046 |
|   5 GCSEs or more | -0.09 (0.20) | -0.47 | .639 |
|   2 A Levels or more | 0.11 (0.19) | 0.56 | .574 |
|   Post-graduate degree | -0.01 (0.21) | -0.07 | .944 |
|  Income (Less than £15,500 as baseline) | | | |
|   £15,500-£24,999 | 0.48 (0.36) | 1.33 | .183 |
|   £25,000-£39,999 | 0.15 (0.31) | 0.49 | .625 |
|   More than £40,000 | -0.04 (0.31) | -0.12 | .903 |
|  Shopping responsibility (half responsibility vs. full) | 0.23 (0.14) | 1.65 | .094 |
|  Income x swap frame interaction | | | |
|   £15,500-£24,999 x social norm frame | 0.09 (0.52) | 0.17 | .862 |
|   £25,000-£39,999 x social norm frame | 0.22 (0.46) | 0.49 | .625 |
|   More than £40,000 x social norm frame | 0.25 (0.44) | 0.56 | .573 |
|   £15,500-£24,999 x cost frame | -0.17 (0.49) | -0.34 | .734 |
|   £25,000-£39,999 x cost frame | 0.49 (0.46) | 1.08 | .279 |
|   More than £40,000 x cost frame | 0.77 (0.45) | 1.72 | .086 |

[a] variables that were controlled for in analyses

## Discussion

This study tested whether framing lower-energy food swaps in terms of cost benefits or social norms would be more effective at encouraging participants to accept such swaps compared to a health message. It was hypothesised that (i) the cost and social norms frames would be more effective than the health frame, and (ii) the cost frame would have a higher impact on swap acceptance for individuals on a low income compared to those on a high income. While offering food swaps led to a significant reduction in energy content of participants' purchases, both at the individual swap level (average reduction 9.69 kcal per swap offer) and at the total basket level (30.48 kcal per basket), this study did not find evidence that one frame was more effective than another at persuading participants to exchange their original high-energy choices for

lower-energy alternatives. This study also did not find evidence of any interaction effects between the type of swap frame and participants' income.

These null effects should be interpreted with caution as there are a number of possible explanations for their emergence. Firstly, the experiment was situated within a simulation online supermarket, and participants were aware that they were not actually purchasing and receiving their chosen products. This could have reduced the impact of emphasising the various benefits of the healthier alternatives to participants, as they knew that they would not really be consuming fewer calories (health frame), saving money (cost frame) or enjoying a product purchased by other consumers (social norms frame). This may have reduced the sensitivity of the experiment for detecting differences between different types of message framing. It is also possible that not enough attention was drawn to the swap messages, or that swap messages were not specific enough as to be salient to participants. For example, social norms messages might be more effective if they emphasise *how many* people preferred the suggested alternative product (e.g., 90% of customers similar to you preferred this product) or by tailoring the message to emphasise common characteristics of the individual to the wider group (e.g., the majority of customers in [LOCATION] preferred this product), as was used in a recent trial to increase uptake of weight management services [41]. A manipulation check to see whether participants recalled which statement they saw would have helped to assess whether low message salience played a part in the null effects, and piloting different messages ahead of implementation could also help determine which phrasing participants most respond to. In addition, the messages framed around social norms may have appeared less credible given that this was not a real shopping site tracking consumers' real purchasing behaviour. Obtaining more detailed feedback from participants on their experience of being offered swaps would help to inform further development of these interventions.

A further limitation is that the products presented in the swap offers may have drawn participants' attention to elements of cost and health regardless of their group allocation, potentially reducing differences between conditions. For example, participants in all groups would have been offered foods clearly labelled as "low fat" or "low calorie", which may have made health salient to participants in the cost and social norms groups as well. Similarly, the healthier alternative products were selected so that they were cheaper than original choices, and while the cost frame group was the only one for whom this information was emphasised in the swap message, all participants would have been able to see the prices of both products (as they would have under real-world circumstances). In addition, participants in all conditions were also asked to stick to a particular budget, which may have made cost information salient to all participants. This could also explain the lack of interaction effects by income group; if all participants were given the same budgetary constraints within the task, then this could have overruled variations in participants' usual attention to cost. Alternatively, the absence of an effect of providing cost incentives could also be due to the small difference in price between original choices and healthier alternatives. The minimum potential price difference between products was £0.01, which may not have been large enough to persuade participants to change their choices.

Alternatively, it is possible that participants in this study did not prioritise cost or social norms any higher than health when making their purchases, meaning that no message frame was any more appealing than the other. Participants may instead have prioritised other factors (e.g., taste, brand familiarity) when making their choices. Participants' priorities for food selection were not measured at any point in this study, meaning that this possibility cannot be tested here. To address both of the above possibilities, future research in this area could include a quick check to see which product features were used by participants to make their choices within the experiment (e.g., familiarity, taste preference, cost, health status etc.) in order to

check which kinds of information participants in different intervention groups pay attention to.

A further possible explanation for the null results in this study is the reduction in sample size due to exclusions. The original planned sample size was 900 participants, which reduced to 713 participants, arguably indicating reductions in power to detect differences between groups. However, this cannot entirely explain these results, particularly because the mean swap acceptance rates show a numeric trend between conditions was in the opposite direction to that expected and accounted for in sample size calculations. The sample size was calculated based on estimated differences in the proportion of accepted swaps, with an anticipated increase in acceptance in the cost and social norms conditions. However, the numeric mean trend shows that acceptance rates were marginally higher in the health frame condition.

Despite no one frame being more effective than another in the current study, there was a significant effect of offering swaps overall. Offering swaps to participants had a significant impact on reducing the calorie content of participants' purchases, both at the individual product level (average reduction 9.69 kcal per product) and at the overall shopping basket level (average reduction 30.48 kcal per basket). However, this is only a relatively small reduction of approximately 0.9% of the energy content of the original baskets chosen by participants. This is likely to be due to the fact that just under 13% of swaps were accepted overall. This is lower than the anticipated 25%, estimated from the study by Forwood and colleagues [21], however this is likely due to the fact that the current study only offered swaps at the checkout, whereas the earlier study compared offering swaps at the checkout versus the point of selection, and found the latter strategy to be more effective. Nevertheless, it is interesting to note that the swap acceptance rate in the present study is more similar to that found in a recent study where 14% of swaps were accepted (when no other interventions were tested concurrently), despite those swaps being offered at the point of selection [23]. Low proportion of swap acceptance has also been cited in earlier research as a contributor to lack of effectiveness for this intervention strategy [21]. Forwood and colleagues found that swaps were more likely to be accepted if they were offered at the point of selection rather than at the checkout [21] which could help to explain the low acceptance rate in our study. However, Koutoukidis and colleagues [23] found that swaps at the point of selection resulted in a similarly low swap acceptance rate, suggesting that other methods of increasing swap acceptance need to be sought if this strategy is to be implemented in practice. One potential improvement could be to enhance the similarity between original choices and healthier alternatives offered in a swap. While the product-matching algorithm that identified healthier swaps selected products from within the same "shelf" category (S2 File), it did not factor in brand or taste proximity, and as a result some swap suggestions were for functionally different foods to the original choice (e.g., salsa dip in exchange for crisps). Other research has found that dissimilarity between original choices and healthier alternatives (i.e., to achieve greater reductions in salt) does not impact likelihood of swap acceptance [22], however it is likely that swaps offering different types of product (salsa versus crisps) would be less acceptable than those offering foods with the same function.

Participant age was significantly associated with swap acceptance, with likelihood of accepting a swap increasing with age, and female gender was associated with a significantly greater reduction in total basket energy content (kcal) from baseline to post-intervention. While these subgroup analyses were exploratory at this stage, this may indicate that different demographic groups are more likely to benefit from this kind of intervention than others. One of the aims of this study was to explore the impact of income level on the effects of the intervention, as an attempt to see whether any evidence for the intervention exacerbating or addressing health inequalities could be found. While no evidence emerged to this effect, a key opportunity for assessing impacts on health inequalities was missed in the absence of recording of participant

ethnicity. It is therefore not possible to assess how well the current sample represents the wider population, and whether these results generalise across ethnic groups. Ethnicity is a key determinant of health outcomes [42], with disparities in obesity rates between different ethnic groups having been recorded. Future research must ensure that interventions are designed and tested with representative samples in order to ensure that they are developed for the populations they aim to benefit.

There are a number of strengths to this study. Firstly, this is the first study to investigate strategies to make food swaps more appealing to consumers. Other researchers have investigated alternative routes to optimising food swap interventions; for example, Payne Riches and colleagues found that online food swap interventions can be optimised by suggesting foods that offer a greater reduction in the nutrient of interest without affecting acceptability or swap acceptance rates [22]. However, no other studies have been conducted that explore how to increase the swap acceptance rate among consumers, and the current study took a particular focus on reducing health inequalities between those on high versus low incomes. Some retailers have already begun to offer healthier food swaps, and so future research should continue to seek opportunities for optimising this intervention to benefit population health.

In addition, these results show that offering lower-energy food swaps in online supermarkets can significantly reduce the energy content of individuals' purchases, even if relatively few of those swaps are accepted. Even small reductions in energy consumed of around 30–100 calories per day could have a significant impact on population health [6] meaning that the present findings should not be discarded. While the acceptance rate in the current study was lower than that observed in some earlier studies [21, 22], it was similar to that found among participants who were offered swaps as a standalone intervention in the study by Koutoukidis and colleagues [23]. Unlike in the present study, these researchers did not find a significant reduction in energy after offering swaps alone [23]. The current findings are promising as they show that a low number of accepted swaps can still have an impact on customers' purchases, which suggests that quick wins could be had if strategies can be found to persuade consumers towards accepting a few more of these swaps. Acceptability of the intervention was relatively high, with participants' mean rating of the intervention being equivalent to stating that they would "somewhat like" to be offered swaps in real life. This dovetails with other findings [22, 23] and suggests that low acceptance rates aren't due to issues of overall acceptability of this approach.

A final strength is that this study found no evidence that such an intervention would contribute to health inequalities, with none of the tested swap frames (or offering swaps overall) influencing one income group over another (although as noted, participants with four GCSEs or fewer were marginally less likely to accept swaps compared to those with a bachelors degree). It should be noted that this is in contradiction to the findings of Forwood and colleagues [21] who found that less deprived participants were more likely to accept swaps. The current findings should also be interpreted with caution, as absence of evidence does not equate to evidence of absence. Future research should continue to monitor whether interventions to influence consumers' shopping behaviour differentially impact people according to their socioeconomic status.

However, there are a number of limitations to this study, indicating a number of avenues for future research. Firstly, asking participants to follow a specified shopping list and to stick to a specific budget may have compromised the task's real-world validity by imposing the requirements of the task over participants' own usual motivations when shopping for groceries. As discussed, it may also have limited the ability to detect differences between income groups as all participants were required to adhere to the same budgetary constraints. It is also possible that these task requirements contributed to participant exclusions; the most common

reason for excluding participants was a final shopping basket totalling fewer than 10 items, which may have been due to the conflict between the items specified on the shopping list, the instruction to only buy items that they would usually buy, and the budgetary constraint. These exclusions resulted in a reduction in sample size which reduced the power of the study to detect the effects under investigation in this study.

A further limitation is that this study did not collect insight on why participants did and did not respond to swap offers. A deeper understanding of the influences underpinning behaviour is crucial for developing behaviour change interventions [43]. As a baseline, it would have been useful to take a measure of participants' priorities for food shopping, with a particular focus on their needs surrounding health, price and popularity (social norms) of foods. This could have helped to understand whether or not the message frames tested here were directed at participants' existing priorities, and also whether price or social norms were indeed more important for their food choices than product healthiness. Future research testing the impact of various message frames should measure the importance participants place on the characteristics highlighted by those frames. In addition, as well as measuring participants' wider priorities, measuring their responses to the individual foods involved in the food swaps could also guide future optimisation attempts. For example, it is possible that the extent to which an originally chosen product is a habitual choice or from a preferred brand influences willingness to exchange it for another.

To conclude, this study found that offering lower-energy swaps can significantly reduce the energy content of consumers' purchases, but that framing the benefits of alternative products in terms of cost or social norms was no more effective than framing in terms of health benefits for increasing swap acceptance rates in a simulation online supermarket. The significant reduction in energy purchased suggests that this is a promising strategy for intervention. However, our findings also show that there is still ample opportunity for improving the appeal of swaps to consumers, and due to the numerous reasons discussed above, the null effects observed here should not be taken as conclusive evidence for abandoning message framing as a strategy. Future research should seek ways to improve the acceptability of food swaps (e.g., by making messages more specific and tailored for consumers), with an ultimate goal to improve consumer health through the online supermarket environment.

## Supporting information

**S1 Table. Results tables for linear regression models on energy change scores for total shopping baskets.**
(DOCX)

**S1 File. Questionnaire materials (Qualtrics).**
(PDF)

**S2 File. List of product categories from within which lower-energy alternatives were identified for swaps.**
(DOCX)

**S1 Data. Data file containing participant survey responses.**
(XLSX)

**S2 Data. Data file containing supermarket data on products included in each swap offer for each participant.**
(XLSX)

**S3 Data. Data file containing supermarket data on products chosen after each swap offer for each participant.**
(XLSX)

## Acknowledgments

The authors are grateful to Rohan Arambepola for his assistance with data processing and analysis, to the Behaviour and Health Research Unit, and to Professor Dame Theresa Marteau for her assistance with the Woods online supermarket platform.

## Author Contributions

**Conceptualization:** Amanda Bunten, Anna Sallis, Sarah Payne Riches, Tim Chadborn, Suzanna Forwood.

**Data curation:** Lucy Porter, Jet G. Sanders, Paul Van Schaik, Marta González-Iraizoz, Suzanna Forwood.

**Formal analysis:** Lucy Porter, Jet G. Sanders, Paul Van Schaik, Marta González-Iraizoz.

**Investigation:** Amanda Bunten, Anna Sallis, Sarah Payne Riches, Suzanna Forwood.

**Methodology:** Amanda Bunten, Anna Sallis, Sarah Payne Riches, Tim Chadborn, Suzanna Forwood.

**Project administration:** Amanda Bunten, Suzanna Forwood.

**Supervision:** Tim Chadborn, Suzanna Forwood.

**Writing – original draft:** Lucy Porter.

**Writing – review & editing:** Amanda Bunten, Lucy Porter, Jet G. Sanders, Anna Sallis, Sarah Payne Riches, Paul Van Schaik, Marta González-Iraizoz, Tim Chadborn, Suzanna Forwood.

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
