## [Decision Letter · Decision Letter 0]

4 Dec 2020

PONE-D-20-32025

A randomised controlled experiment of health, cost and social norm message frames to encourage acceptance of food swaps in a virtual supermarket

PLOS ONE

Dear Dr. Porter,

Thank you for submitting your manuscript to PLOS ONE. After careful consideration, we feel that it has merit but does not fully meet PLOS ONE’s publication criteria as it currently stands. Therefore, we invite you to submit a revised version of the manuscript that addresses the points raised during the review process.

Both reviewers think the paper is interesting. However, they also point out the two major problems. 

1. Inadequate sample size. 

2. The lack of control in the experiment. 

I hope you can sufficiently address these comments in your revision. Please also reply to the reviewers' comments item by item following the instruction below. 

We look forward to receiving your revised manuscript.

Kind regards,

Zhifeng Gao

Academic Editor

PLOS ONE

Journal Requirements:

2. Please consider modifying your title to ensure that it is specific, descriptive, concise, and comprehensible to readers outside the field, by specifying that the study used a simulation approach. This should also be clear in the Abstract.

3.In your Data Availability statement, you have not specified where the minimal data set underlying the results described in your manuscript can be found. PLOS defines a study's minimal data set as the underlying data used to reach the conclusions drawn in the manuscript and any additional data required to replicate the reported study findings in their entirety. All PLOS journals require that the minimal data set be made fully available. For more information about our data policy, please see http://journals.plos.org/plosone/s/data-availability.

4. Please ensure that you refer to Figure 1 in your text as, if accepted, production will need this reference to link the reader to the figure.

Reviewers' comments:

Reviewer's Responses to Questions

**Comments to the Author**

1. Is the manuscript technically sound, and do the data support the conclusions?

Reviewer #1: Partly

Reviewer #2: Yes

2. Has the statistical analysis been performed appropriately and rigorously? 

Reviewer #1: Yes

Reviewer #2: Yes

3. Have the authors made all data underlying the findings in their manuscript fully available?

Reviewer #1: Yes

Reviewer #2: Yes

4. Is the manuscript presented in an intelligible fashion and written in standard English?

Reviewer #1: Yes

Reviewer #2: Yes

5. Review Comments to the Author

Reviewer #1: I enjoyed reading this very interesting article on framing effect and food swap. The article addresses a very practical issue on how to “nudge” people for a last-minute healthy alternative. The manuscript is comprehensive and well written.

As the authors pointed out in the introduction and discussion, this is a hypothetical experiment conducted online. The authors attempted to compare cost frame and social norm frame to the health frame. My major concern is that the experiment does not have a control group. What is inherent in a health frame is unknown, therefore the difference between a health frame and a cost frame is implicit. The same goes for the social normal frame. Is being healthy considered as part of the social norm? without a true control group that has no framing at all, it is hard to interpret the findings. This is perhaps why the authors struggled with explaining the null effect - because it is not really a null effect, it is a health effect. Since the experiment was conducted online, the authors should be able to collect another control group, bearing that the new sample is random and independent from the three framing groups. If for some reason, the authors are unable to collect a control group, analysis plan should be adjusted to use descriptive statistics and statistical tests to analyze the difference among the three groups, rather than pooling all data in a regression framework.

Minor

The motivation for presenting models 1-3 is unclear. Unless you are using a different dependent variable in each model, you should only present the preferred model.

In table 2, there seem to be some difference in education between the three conditions. A statistical test should be conducted on the differences.

Thank you for the opportunity to learn about your work.

Reviewer #2: This is the second time I am reviewing this paper. I am pleased that the authors have taken most of my previous comments into consideration. I hope the comments below can help strengthen the manuscript.

The main limitation of the manuscript are that the proposed sample size was not reached and therefore we do not know if there was actually evidence of no effect or no evidence of effect. However, it is an otherwise well-conducted and well-reported study that can add to the literature of how to assess potential new interventions to promote healthier choices in online supermarkets. Such studies are urgently needed, particularly given the change in shopping practices and dietary habits due to COVID-19.

Abstract

Reword kj/kcal to kJ or kcal, as it may cause confusion.

The presentation of the results in the abstract does not align with the randomised design. For example, you need to first present the results by trial group and then specify that when all groups were combined in a pre-post design there was an indication of effect.

“Predictor” has very specific implications, rewording this to moderator throughout is suggested. Also specifying that this was exploratory.

Background

Well written, logic flow, and makes the case for the need for this study.

Line 54: Suggest you add some non-UK data too to appeal to an international audience.

Methods

Line 109: Delete “was”.

Line 114: Did you take into account dropouts in the sample size calculation?

Line 123: More detail is needed re the method and process of randomisation and allocation concealment. See CONSORT guidance.

Line 151: Were the two swaps appeared simultaneously (ie. Next to each other) or did the second only appear after the first was rejected.

Please add information on whether you considered other characteristics (e.g. taste proximity, brand) between offered product to the originally selected, as a same-brand low-kcal cheddar cheese might be much more acceptable than a random one.

It would be extremely useful for the readers to grasp the intervention if Table 1 was replaced by an example snapshot of the online supermarket while swaps were offered.

Line 190: “was planned to” might imply that this was an original plan but what happened in reality differed. Worth clarifying.

Lines 194, 224 and throughout: I assume authors mean sex rather than gender?

Line 210: Suggest reword analysis (ii) to “effect of swap frame on energy reduction”

Line 290: Reword “was” to “would have been”

Worth adding p-values for comparisons in Figure 1.

Discussion

Well-balanced.

I would add in the cost paragraph that potentially the lack of minimal cost cut-off might have contributed to no effect. It is possible that if something was simply £0.01 cheaper, people wouldn’t want to sacrifice their original choice for such a small benefit.

6. PLOS authors have the option to publish the peer review history of their article (what does this mean?). If published, this will include your full peer review and any attached files.

Reviewer #1: No

Reviewer #2: **Yes: **Dimitrios Koutoukidis

---

## [Author Response · Author response to Decision Letter 0]

13 Jan 2021

(Note: we have also uploaded this response as a file for ease of reading re: distinguishing between reviewer advice and our response.)

Dear Dr. Gao,

Thank you very much for giving us the opportunity to respond to the reviewers’ comments.

We have responded to each of them below and hope that the amendments we have made will meet your requirements.

Response to Reviewer #1

I enjoyed reading this very interesting article on framing effect and food swap. The article addresses a very practical issue on how to “nudge” people for a last-minute healthy alternative. The manuscript is comprehensive and well written.

As the authors pointed out in the introduction and discussion, this is a hypothetical experiment conducted online. The authors attempted to compare cost frame and social norm frame to the health frame. My major concern is that the experiment does not have a control group. What is inherent in a health frame is unknown, therefore the difference between a health frame and a cost frame is implicit. The same goes for the social normal frame. Is being healthy considered as part of the social norm? without a true control group that has no framing at all, it is hard to interpret the findings. This is perhaps why the authors struggled with explaining the null effect - because it is not really a null effect, it is a health effect. Since the experiment was conducted online, the authors should be able to collect another control group, bearing that the new sample is random and independent from the three framing groups. If for some reason, the authors are unable to collect a control group, analysis plan should be adjusted to use descriptive statistics and statistical tests to analyze the difference among the three groups, rather than pooling all data in a regression framework.

Thank you very much for your comments, we are glad that you found the manuscript comprehensive and well-written. In response to the concern regarding the Control group, we agree that the Active Control group (i.e., the health frame) used in this study is conceptually different from the idea of a “no frame” Control group that offers swaps without an accompanying message. However, the research question of focus for this study was whether framing swaps in terms of cost or social norms was a more effective strategy than framing in terms of health benefits. Given this, using the health frame group as a comparison group is the most appropriate strategy for answering this question.

Nevertheless, your comments encouraged us to think about this issue more broadly and we have made a couple of amendments to reflect this, which we hope you will approve of. Firstly, we have reconceptualised the description of the study as a “randomised experiment” (rather than a randomised controlled experiment) to reflect the fact that the Control group in this study still involved exposure to an intervention. 

Secondly, after reflecting on your comment about what is inherent in a health/social norm/cost frame, we arrived at the conclusion that in this experiment, even a no-frame condition could be considered as inherently health-related. This is because the products offered in exchange always had a lower energy content, and while this may not have always been apparent (e.g., when one type of “full-fat/full-sugar” sweet was suggested as a swap for another full-fat/sugar sweet), in a number of instances, participants would have been presented with an option that was evidently lower energy (e.g., due to branding indicating “light” products). This would have been the same across conditions, regardless of whether the message frame drew attention to the energy content of the product. We have therefore added this to the discussion as a limitation. 

The motivation for presenting models 1-3 is unclear. Unless you are using a different dependent variable in each model, you should only present the preferred model.

Thank you for your comment – we have reflected on this and have decided to remove model 3 as this was an exploratory question and was not specified in our pre-registration. Model 2 (i.e., that adjusted for demographic variables) is considered our preferred model, however we have also included the unadjusted model of intervention group alone (i.e., Model 1) as standard. The models are now referred to as the “adjusted” and “unadjusted” models, which we hope will aid understanding.

In table 2, there seem to be some difference in education between the three conditions. A statistical test should be conducted on the differences. Thank you for the opportunity to learn about your work.

Thank you for your comment – we had originally conducted these tests for all of our baseline characteristics, however an earlier reviewer instructed us to remove these, citing the CONSORT guidelines and the paper by de Boer, Waterlander, Kuijper, Steenhuis and Twisk (2015) which makes the case for not doing so. Given these guidelines, we have not included them in the manuscript but our earlier test found that the difference in education was not significant across conditions (p = .774).

Thank you very much for taking the time to review our work. We hope that the amendments made are to your satisfaction.

Response to Reviewer #2

This is the second time I am reviewing this paper. I am pleased that the authors have taken most of my previous comments into consideration. I hope the comments below can help strengthen the manuscript.

The main limitation of the manuscript are that the proposed sample size was not reached and therefore we do not know if there was actually evidence of no effect or no evidence of effect. However, it is an otherwise well-conducted and well-reported study that can add to the literature of how to assess potential new interventions to promote healthier choices in online supermarkets. Such studies are urgently needed, particularly given the change in shopping practices and dietary habits due to COVID-19.

Thank you for your comments, and for taking the time to review our paper for the second time. We agree that the reduced sample size is a limitation for the manuscript, and we have now emphasised this more in our discussion. Your comment encouraged us to reflect further on this point and we believe that lowered power cannot entirely explain our results as the mean trend (while non-significant) showed a pattern in the opposite direction to the one we expected. As a result, we believe that other factors (such as the fact that all participants were asked to stick to a budget, or the fact that all participants could see price information) may be more important for the null results observed here. We have emphasised this point too. We hope that the methodological learnings from this paper can be of use for others developing and evaluating interventions in the field, and believe that our paper makes a useful contribution in this regard.

Abstract

Reword kj/kcal to kJ or kcal, as it may cause confusion.

Thank you, we have reworded this.

The presentation of the results in the abstract does not align with the randomised design. For example, you need to first present the results by trial group and then specify that when all groups were combined in a pre-post design there was an indication of effect.

Thank you, we have now reordered the presentation of results.

“Predictor” has very specific implications, rewording this to moderator throughout is suggested. Also specifying that this was exploratory.

Thank you for your suggestion, we don’t believe that moderator is the right term for this relationship as (for example, in this case) the age � swap acceptance analysis was a main effect rather than an interaction. We have reworded to describe “associations” instead.

Background

Well written, logic flow, and makes the case for the need for this study.

Line 54: Suggest you add some non-UK data too to appeal to an international audience.

Thank you, we have added some non-UK data to describe global shopping habits, particularly contextualised within the coronavirus pandemic.

Methods

Line 109: Delete “was”.

This has been done.

Line 114: Did you take into account dropouts in the sample size calculation?

No, dropouts were not accounted for. This is now clearer in the manuscript.

Line 123: More detail is needed re the method and process of randomisation and allocation concealment. See CONSORT guidance.

Thank you for highlighting this, we have now provided more information on the randomisation allocation ratio and allocation concealment.

Line 151: Were the two swaps appeared simultaneously (ie. Next to each other) or did the second only appear after the first was rejected.

Both appeared simultaneously. We have now amended this to make it clearer.

Please add information on whether you considered other characteristics (e.g. taste proximity, brand) between offered product to the originally selected, as a same-brand low-kcal cheddar cheese might be much more acceptable than a random one.

Thank you, more information has been added to the manuscript. To summarise in brief, the algorithm did not consider characteristics such as taste proximity and brand. Alternative products were selected from within the same shelf. Unfortunately, this means that some swaps were not “in kind” (e.g., a pot of salsa being suggested as a swap for crisps). We have added this as a limitation to the discussion.

It would be extremely useful for the readers to grasp the intervention if Table 1 was replaced by an example snapshot of the online supermarket while swaps were offered.

Thank you – we do not have screenshots of the intervention, however we have created mock-ups for readers to see.

Line 190: “was planned to” might imply that this was an original plan but what happened in reality differed. Worth clarifying.

Thank you, we have reworded.

Lines 194, 224 and throughout: I assume authors mean sex rather than gender?

The wording of the questionnaire that participants saw was gender, and so this term is retained.

Line 210: Suggest reword analysis (ii) to “effect of swap frame on energy reduction”

Line 290: Reword “was” to “would have been”

Worth adding p-values for comparisons in Figure 1.

Discussion

Well-balanced.

I would add in the cost paragraph that potentially the lack of minimal cost cut-off might have contributed to no effect. It is possible that if something was simply £0.01 cheaper, people wouldn’t want to sacrifice their original choice for such a small benefit.

Thank you for the suggestions, we have amended as advised.

Thank you again for your time and contributions to this manuscript on its second round, we hope that the amendments have been made to your satisfaction.

---

## [Decision Letter · Decision Letter 1]

20 Jan 2021

A randomised experiment of health, cost and social norm message frames to encourage acceptance of swaps in a simulation online supermarket

PONE-D-20-32025R1

Dear Dr. Porter,

We’re pleased to inform you that your manuscript has been judged scientifically suitable for publication and will be formally accepted for publication once it meets all outstanding technical requirements. **However, reviewers and I both noticed some typos, please carefully check your final manuscript to make corrections. **

Kind regards,

Zhifeng Gao

Academic Editor

PLOS ONE

Additional Editor Comments (optional):

Reviewers' comments:

Reviewer's Responses to Questions

**Comments to the Author**

1. If the authors have adequately addressed your comments raised in a previous round of review and you feel that this manuscript is now acceptable for publication, you may indicate that here to bypass the “Comments to the Author” section, enter your conflict of interest statement in the “Confidential to Editor” section, and submit your "Accept" recommendation.

Reviewer #1: All comments have been addressed

Reviewer #2: All comments have been addressed

2. Is the manuscript technically sound, and do the data support the conclusions?

Reviewer #1: Yes

Reviewer #2: Yes

3. Has the statistical analysis been performed appropriately and rigorously? 

Reviewer #1: Yes

Reviewer #2: Yes

4. Have the authors made all data underlying the findings in their manuscript fully available?

Reviewer #1: Yes

Reviewer #2: Yes

5. Is the manuscript presented in an intelligible fashion and written in standard English?

Reviewer #1: Yes

Reviewer #2: Yes

6. Review Comments to the Author

Reviewer #1: This revision has significantly improved based on my comments. There are some typos in the manuscript. Please check carefully based on the Journal's format guide.

Reviewer #2: Thank you for the revised manuscript, my comments have been addressed. I have nothing further to add.

7. PLOS authors have the option to publish the peer review history of their article (what does this mean?). If published, this will include your full peer review and any attached files.

Reviewer #1: No

Reviewer #2: **Yes: **Dimitrios Koutoukidis

---

## [Editor Report · Acceptance letter]

26 Jan 2021

PONE-D-20-32025R1 

A randomised experiment of health, cost and social norm message frames to encourage acceptance of swaps in a simulation online supermarket 

Dear Dr. Porter:

I'm pleased to inform you that your manuscript has been deemed suitable for publication in PLOS ONE. Congratulations! Your manuscript is now with our production department. 

Kind regards, 

on behalf of

Dr. Zhifeng Gao 

Academic Editor

PLOS ONE